



# An extensive database of airborne trace gas and meteorological observations from the Alpha Jet Atmospheric eXperiment (AJAX)

Emma L. Yates[1,2], Laura T. Iraci[1], Susan S. Kulawik[1,2], Ju-Mee Ryoo[1,3], Josette E. Marrero[1,†], Caroline L. Parworth[1,‡], Jason M. St. Clair[4,5], Thomas F. Hanisco[4], Thao Paul V. Bui[1], Cecilia S. Chang[1,2], Jonathan M. Dean-Day[1,2]

[1]NASA Ames Research Center, Moffett Field, CA 94035, USA

[2]Bay Area Environmental Research Institute, Moffett Field, CA 94035, USA

[3]Science and Technology Corporation, Moffett Field, CA 94035, USA

[4]Atmospheric Chemistry and Dynamics Laboratory, NASA Goddard Space Flight Center, Greenbelt, MD 20771, USA

[5]Joint Center for Earth Systems Technology, University of Maryland Baltimore County, Baltimore, MD 21228, USA

†Now at Sonoma Technology, Inc., Signal Hill, CA 90755, USA

‡Now at Aclima, Inc., San Francisco, CA 94111, USA

*Correspondence to*: Emma L. Yates (emma.l.yates@nasa.gov) and Laura T. Iraci (laura.t.iraci@nasa.gov)

**Abstract.** The Alpha Jet Atmospheric eXperiment (AJAX) flew scientific flights between 2011 and 2018 providing measurements of trace gas species and meteorological parameters over California and Nevada, USA. This paper describes the observations made by the AJAX program over 229 flights and approximately 450 hours of flying. AJAX was a multi-year, multi-objective, multi-instrument program with a variety of sampling strategies resulting in an extensive dataset of interest to a wide variety of users. Some of the more common flight objectives include satellite calibration/validation (GOSAT, OCO-2, TROPOMI) at Railroad Valley and other locations, and long-term observations of free-troposphere and boundary layer ozone allowing for studies of stratosphere-to-troposphere transport and long-range transport to the western United States. AJAX also performed topical studies such as sampling wildfire emissions, urban outflow, and atmospheric rivers. Airborne measurements of carbon dioxide, methane, ozone, formaldehyde, water vapor, temperature, pressure, and 3-D winds made by the AJAX program have been published at NASA's Airborne Science Data Center.

## 1 Introduction

The Alpha Jet Atmospheric eXperiment (AJAX) was a flight measurement program that utilized a tactical military aircraft modified for for scientific observations (see Figure 1, left). The AJAX program was the result of a public-private partnership between NASA Ames Research Center (NASA ARC) and H211 LLC, which owns the Alpha Jet. This partnership provided



the opportunity to acquire scientific observations from 2011 to 2018, taking measurements of trace gas species and meteorological parameters over California and Nevada, USA. It has produced a unique multi-year, multi-objective, multi-instrument airborne dataset, with 229 flights and ~450 flight hours. The distribution of AJAX flights by year and season, along with common flight sampling locations are shown in Figure 1.

The AJAX program flew on average 2-3 science flights a month, as determined by aircraft availability and weather conditions. Flight opportunities were typically arranged the week prior to a science flight, in which consideration of synoptic conditions, time of year and science objectives determined desired flight objectives and target locations. This flexibility and rapid turnaround meant that the AJAX program could respond to short-lived regional scientific opportunities, such as sampling emissions from wildfires or atmospheric rivers. This flexibility also allowed for the participation in, and collaboration with, more traditional field campaigns (e.g., CABOTS, LVOS, DISCOVER-AQ-CA, SEAC4RS). Table 1 lists some of the more common AJAX flight objectives, definitions, and number of flights associated with each objective; note that many AJAX flights accomplished multiple objectives in a single flight.

Given the wide variety of flight objectives shown in Table 1, AJAX data have been used to investigate a variety of science questions including:

    i. Satellite validation: AJAX regularly sampled vertical profiles of the atmosphere from ~8 km to the lowest safe altitude within the boundary layer (1000 ft in congested (urban) areas and 500 ft over sparsely populated regions (FAR, 2023)). For example, the regularly-used satellite validation site, Railroad Valley (RRV), NV is a common AJAX flight target location. The AJAX program has a long record of vertical profile observations of $CO_2$ and $CH_4$ in conjunction with satellite overpasses and has been used as comparison data for GOSAT (Tanaka et al., 2016), GOSAT-2 and OCO-2 (Kulawik et al., 2017). In addition, AJAX flights also performed vertical profiles of ozone and formaldehyde under TROPOMI, and these data are still to be fully analyzed (Parworth et al., 2018).

    ii. Free tropospheric ozone: Free tropospheric ozone trends over the western US has been an active area of research for many years. Data provided by flights from the AJAX program have proved to be a useful asset in assessing ozone trends and identifying extreme events, such as stratosphere-to-troposphere transport and long-range transport of pollution (e.g., Faloona et al., 2020; Fine et al., 2015; Langford et al., 2018; Ryoo et al., 2017; Yates et al., 2013). The long-term, sustained observational approach adopted by the AJAX program is of value in assessing changes in free-troposphere ozone trends (Chang et al., 2022; Lin et al., 2015).

    iii. Fires: AJAX has sampled a total of 14 wildfires over 17 different flights, with some fires sampled more than once (Iraci et al., 2021b). Iraci et al. (2022) provides an overview of wildfire emissions sampled by AJAX. In addition, there have been focused analyses of emissions from California's Rim wildfire during 2013 and Soberanes fire during 2016 as reported by Baker et al. (2018), Yates et al. (2016) and Langford et al. (2020).

    iv. Urban and point source emissions: The Alpha Jet's fast cruising speeds can be useful in reducing uncertainties due to the evolution of the boundary layer during flight sampling. AJAX can essentially take a "snapshot" of the boundary layer and has been used to assess outflow of $CO_2$ and $CH_4$ emissions from urban centers (Ryoo et al.,



2019), and emission point sources (Leifer et al., 2018; Tadić et al., 2017), as well as the transport of emissions (Leifer et al., 2020).

v. Atmospheric rivers: The flexibility of the AJAX program was an advantage to provide a rapid response to measure the early phase of atmospheric river events as demonstrated by Ryoo et al. (2020). AJAX examined terrain-trapped airflows during an atmospheric river event which impacted northern California in March 2016, providing spatially resolved 3-D wind observations along a portion of the coastline where wind profiler data did not exist.

vi. Central Valley: AJAX collaborated with NASA's DISCOVER-AQ-CA field campaign in 2013 to sample the boundary layer in California's San Joaquin Valley, a region with some of the most severe ozone and PM2.5 pollution in the U.S. AJAX ozone, $CO_2$ and $CH_4$ data have been used for observational and model studies of air quality and greenhouse gas emissions in this region (e.g. Faloona et al., 2020; Cui et al., 2019; Johnson et al. 2014; Johnson et al. 2016; Yates et al., 2015) as well as the impacts on surrounding, rural areas (Yates et al., 2020).

This paper provides a detailed overview of the AJAX program and its data products. We present a summary of the range of flight objectives, sampling strategies, airborne instrumentation, and data processing to produce AJAX trace gas and meteorological data products. An overview of the AJAX dataset is presented, including discussions on long-term trends and greenhouse gas correlations and a case study highlighting the use of AJAX data in support of satellite validation of lowermost tropospheric products from OCO-2 and GOSAT.

## 2 Sampling strategy

AJAX flights were typically 2 hours in duration, starting and ending at Moffett Field, CA (37.415 °N, 122.050 °W); take-off times were generally between 10:00 and 12:00 local time. Horizontally the measurements covered an area of 32° to 42° N and -125° to -115 ° W (Figure 1). Flight patterns varied considerably depending on the flight objective (see Table 1), which was determined ~1 week prior to the flight date, based on weather, science, and aircraft availability. Some of the more common AJAX flight patterns are discussed below.

## 2.1 Tropospheric profiling

AJAX has a long-term record of designing and executing science flights to perform tropospheric profiling, often coincident with satellite overpasses, a Total Carbon Column Observing Network (TCCON) site, or ozonesonde launches from Trinidad Head, CA (THD). The sustained, long-term approach by AJAX allows for validation of remotely-sensed data products during the evolution of the instrument/mission lifetime, providing a unique dataset and a powerful asset for continued data evaluation. In total, vertical profiles of trace gases (≥8 km to boundary layer) have been recorded in 127 times in the AJAX data record, including 58 flights over RRV with the primary purpose of collecting $CO_2$ and $CH_4$ measurements for validation of GOSAT, GOSAT-2 and OCO-2 satellite data products (Tadić et al., 2014; Tanaka et al., 2016). In addition, AJAX flights targeting offshore vertical profiles coincident with OCO-2 provided data aiding the development of a lowermost tropospheric $CO_2$ partial





column satellite product (Kulawik et al., 2017). AJAX has also collected trace gas vertical measurements coincident with TROPOMI and a TCCON site based at NASA Armstrong Flight Research Center (NASA AFRC), formerly known as Dryden

Flight Research Center.

The sampling strategy for a tropospheric profiling flight generally followed this pattern: ascent from Moffett Field reaching a cruise altitude of ~8 km shortly after take-off for transit to a desired location, followed by a descending, spiraling profile with a descent rate of 10 m/s and horizontal diameter of <9 km The bottom of the profile can be at or below 50 m above ground level in accordance with FAR 91.119 in sparsely populated areas (FAR, 2023). After completion of the vertical profile, AJAX

typically ascended and returned to Moffett Field with the return transit leg at a lower altitude (e.g., 5 km) than the outbound transit leg. AJAX observations of $CO_2$ and $CH_4$ over RRV measured during descending profiles are presented in Figure 2. Annual increases in $CO_2$ and $CH_4$ are observed over this long-term record, as discussed further in Section 4.

**2.2 Boundary layer observations**

AJAX has a record of regular, low-altitude flights observing the San Francisco Bay Area (SFBA), California's Central Valley

(CCV) and offshore boundary layers. AJAX take-off and landing data provide a long-term record of the SFBA boundary layer, which has currently been underused. AJAX executed regular flights with the main objective to measure within the SFBA (13 flights), CCV (85 flights) and offshore (14 flights) boundary layers to better understand emission sources and transport.

The sampling strategy of boundary layer flights was typically: take-off from Moffett Field and transit towards the first desired location at an altitude left to pilots' discretion. On arrival at the first sampling location AJAX would descend to ≤300 m and

from there would sample at altitudes ≤300 m through several designated locations within the study region before transiting and returning to Moffett Field, providing ~1.5 hours of boundary layer observations within a 2-hour flight. An example AJAX flight focusing on boundary observations is shown in Figure 3, which displays $O_3$, $CH_4$ and $CO_2$ collected on one day of a mini-intensive coordinated during a seasonal deployment of multiple EM27/SUN ground-based spectrometers across the San Francisco Bay Area of California (Klappenbach et al., 2021). The flight plan was devised to provide in-situ greenhouse gas

measurements near and above multiple column observations on sequential days in Fall 2016.

**2.3 Source identification**

AJAX had several flight objectives to sample and help quantify emissions sources, including wildfires (17 flights), emissions from urban centers (14 flights) and oil and gas infrastructure (17 flights). These flights required a sampling approach that firstly aims to obtain an upwind, clean-air sample before targeting observations from the emission source itself. A typical

sampling strategy would be as follows: take-off from Moffett Field, ascend to a transit altitude (pilot discretion), perform a vertical profile descending from ≤ ~8 km to within the boundary layer, upwind as close to the emission source as feasible. AJAX would then perform several circles or passes around the emission source within the boundary layer. Often several altitudes were flown, allowing for 2-3 circles around the source before ascending for the return transit (altitude at pilot's discretion) and landing at Moffett Field. An example source identification AJAX flight is shown in Figure 4 from a flight over



the Aliso Canyon natural gas leak in December 2015. The observed, uncorrelated $CO_2$ and methane are due to difference in emission sources. High $CO_2$ was observed downwind of Aliso Canyon and is associated with transport of pollution from the LA basin, whereas extremely high methane values are observed in the vicinity of Aliso Canyon.

## 3 Description of airborne platform, sensors, and data processing

The Alpha Jet has four external wing-pods; the two outboard pods on each wing contain fuel, and the two-inboard pods (painted
white in Figure 1) are redesigned fuel tanks which can house scientific instruments, as shown in Figure 5. The instrument wing-pods can each carry a payload of 245 kg within 0.15 $m^3$ of available space, and modifications in the rear cockpit allow for electrical (on/off) control of the instrument payload. A NASA Airworthiness Statement was issued following thorough Airworthiness and Flight Safety Reviews of the scientific instrumentation and planned operations. During all science flights the Alpha Jet operated as a public-use aircraft under a flight release issued by NASA Ames Research Center.

The AJAX payload was developed incrementally. AJAX flew its first science flight in January 2011 with an ozone ($O_3$) monitor. In June 2011 a greenhouse gas monitor was added, measuring carbon dioxide ($CO_2$), methane ($CH_4$) and water vapor ($H_2O$). In June 2013, the Meteorological Measurement System (MMS) was added to measure 3-D winds, temperature, and pressure, and lastly the addition of the Compact Formaldehyde Fluorescence Experiment (COFFEE) was completed in December 2015.

The Alpha Jet flies at speeds between 75-230 m/s with a ceiling height of 12 km, however science flights were typically flown at speeds of ~140 m/s (~500 km/hr) up to an altitude of 8 km (limited by decreasing performance of the greenhouse gas instrument at higher altitudes).

### 3.1 Ozone monitor

AJAX observes $O_3$ mixing ratios using a commercial ozone monitor (2B Technologies Inc., model 205, S/N 734), modified
for flight worthiness and improved performance. Details of the instrument have been reported by Yates et al. (2013). Modifications of the $O_3$ monitor include an upgrade of the pressure sensor and pump to allow measurements at high altitudes (low pressures), the inclusion of a lamp heater to improve the stability of the UV source, and the addition of heaters, temperature controllers and vibration isolators to control the monitor's physical environment. The air intake is through Teflon tubing (perfluroalkoxy-polymer, PFA) with a backward-facing inlet positioned on the underside of the instrument wing-pod.
Air is delivered through a 5 μm PTFE (polytetrafluroethylene) membrane filter to remove fine particles prior to analysis.

The $O_3$ monitor has undergone thorough instrument testing in the laboratory and in a pressure- and temperature- controlled environment to determine the precision, linearity, and overall uncertainty. Eight-point calibration tests (ranging from 0 – 300 ppbv) were performed before and/or after most AJAX flights using an $O_3$ calibration source (2B Technologies, model 306, S/N 045, referenced to the WMO scale). The uncertainty of AJAX $O_3$ measurements is estimated from the sum in quadrature
of the laboratory precision (1-σ over 3 min at 10 s sampling resolution), variability in laboratory calibrations (the differences



observed in the calculated zero offset when sampling the ozone source set to 0 ppb), repeatability (maximum difference in ozone observed from three calibrations on the same day) and pressure dependance (the change in ozone observed with pressure changes while sampling 50 ppb ozone in environmental chamber tests). The overall uncertainty of the airborne AJAX $O_3$ data are estimated to be ~3.0 ppbv, based on laboratory and chamber testing; a breakdown of the uncertainty terms is shown in Table 2.

AJAX performed regular flights to Trinidad Head (THD) in conjunction with ozonesonde launches for comparison and validation of AJAX $O_3$ data and to the Table Mountain Facility (TMF) for comparison with the TMF tropospheric ozone lidar (TMTOL) and TMF ozonesonde. Figure 6 shows an example of coincident ozone profiles taken over TMF on 24 May 2013 by AJAX, TMTOL and TMF ozonesonde. Comparisons show very good agreement between the airborne, TMTOL and TMF ozonesonde measurements, well within measurement uncertainties, despite spatial and temporal differences in data sampling.

To provide quality controlled $O_3$ data, we applied standard processing to the raw data as follows:

— Removal of datapoints outside of a predefined operational limit. If the reported flow rate through the sample cell fell below 1 L/min, those datapoints were removed from the dataset.

— Removal of outliers in the data due to instrument instability. If $\Delta O_3$ ($O_3$ (i)- $O_3$ (i+1)) is greater than the 1σ standard deviation of the entire $O_3$ data, those datapoints ($O_3$(i) and $O_3$(i+1)) were removed from the dataset.

— Calibration based on the linearity and zero-offset factors calculated from the closest calibration to the flight (typically ±1 day).

—  Averaging of $O_3$ data from 2 s resolution to report at 10 s resolution, improving precision and the overall quality of the finalized dataset.

## 3.2 Greenhouse gas sensor

AJAX provides in situ measurements of $CO_2$ and $CH_4$ using a commercial sensor (Picarro Inc. G2301-m, S/N 634-CFDDS2120) which has been modified for flight. Details of the sensor have been reported elsewhere (Tadic et al., 2014, Tanaka et al., 2016). Briefly, the AJAX GHG sensor has been re-packaged into two separate enclosures (analyzer and electronics) to fit within the wing-pod. Additional modifications to the original sensor for flight readiness included fans for thermal management, an additional filter on the inlet, and vibration isolators. The GHG sensor is powered on ~1 hr prior to flight take-off in a pre-flight routine that ensures the laser temperature reaches its operational temperature of 45 °C and commences measurements early into each science flight.

The GHG sensor has undergone thorough laboratory and environmental chamber testing to evaluate its precision, linearity, and overall accuracy under a variety of operating conditions. Calibrations are performed before and/or after most AJAX flights using NOAA ERSL whole air standards certified by the WMO Central Calibration Laboratory for $CO_2$ and $CH_4$. The overall uncertainty of aircraft measurements was estimated considering the precision (1σ), repeatability of calibrations, in-flight variance due to cavity pressure fluctuations, uncertainty in water vapor corrections, and pressure dependence of measurements based on environmental chamber studies.



AJAX $CO_2$ and $CH_4$ measurements were subjected to the following quality control procedures to generate the finalized GHG

dataset:

— Removal of instrument lag time (5 s), calculated given the length of the inlet tubing and measured flow rate.

— Removal of outliers in the data due to in-flight instability of the optical cavity pressure. Datapoints where the cavity pressure deviated by more than 0.2% were removed from the dataset.

— Removal of datapoints outside of operational temperature limits. Datapoints were removed when the cavity

temperature was >45.15 °C or <44.95 °C.

— Water vapor corrections. AJAX applies the water vapor corrections described by Chen et al. (2010) using simultaneous water observations.

— Application of calibration based on the closest calibration to the flight (typically ±1 day).

— Averaging of GHG data from 3 Hz resolution to report at 3 s resolution, improving precision and the overall quality

of the reported, finalized dataset.

Typical overall uncertainties for AJAX $CO_2$ and $CH_4$ are 0.16 ppmv and 2.2 ppbv respectively as shown in Table 3 and reported by Tanaka et al. (2016).

### 3.3 Formaldehyde instrument: COmpact Formaldehyde FluorescencE Experiment

AJAX collected in-situ observations of formaldehyde (HCHO) using a custom-built instrument called COmpact Formaldehyde

FluorescencE Experiment (COFFEE), which was designed specifically to join the AJAX payload. A detailed overview of COFFEE is presented by St. Clair et al. (2017), with updates to the optical design in St. Clair et al. (2019). Briefly, COFFEE utilizes non-resonant laser-induced fluorescence (NR-LIF) to measure formaldehyde, with 300 mW of 40 kHz 355 nm laser output exciting multiple formaldehyde absorption features and the resulting fluorescence measured by two photomultiplier tube detection axes. Fluorescence signal is collected for 500 ns, with 5 ns resolution, at each laser pulse and then averaged to

1 s. The 1 Hz fluorescence signal shape is processed post-flight to calculate formaldehyde mixing ratios.

Data collected prior to April 2018 (prior to the optical changes described in St. Clair et al., 2019) have a water dependence term in the measurement uncertainty that is not present in data collected afterward. Overall measurement uncertainty is ± (20% of HCHO + 200 pptv + 230*[$H_2O$ % by volume]) before April 2018 and ± (20% of HCHO + 200 pptv) in April 2018 and after.

### 3.4 Meteorological Measurement System (MMS)

AJAX reports calibrated, science-quality static pressure, static temperature, and 3-dimensional winds using the NASA-Ames developed Meteorological Measurement System (MMS). MMS consists of three major systems: An air motion sensing system, an inertial navigation system and a data acquisition system. MMS has been deployed on multiple NASA aircraft and has been involved in numerous NASA field campaigns. A detailed overview of MMS instrumentation and basic concepts are described by Scott et al. (1990).



The primary products of MMS are pressure (precision of ±0.3 mb with accuracy of 0.5%), temperature (±0.3 K, 0.2%), horizontal wind (±1m/s, ~3.3%) and vertical wind (±0.3 m/s). The derived parameters are potential temperature, true air speed, turbulence dissipation rate and Reynolds number. MMS reporting parameters are GPS positions, velocities, accelerations, pitch, roll, yaw, heading, angle of attack, angle of sideslip, dynamic total pressure, and total temperature.

## 4 Dataset overview

For each AJAX flight, four individual instrument datasets are available for download following the ICARTT file format V2.0. The $CO_2$ + $CH_4$ dataset (3 s resolution) and $O_3$ dataset (10 s resolution) each include time and aircraft position. The HCHO dataset is available at 1 s and includes a time stamp; GPS data can be inferred from either the MMS or greenhouse gas datasets. The MMS dataset includes meteorological parameters and aircraft position archived at 1 s. All AJAX Level 2 datasets (https://doi.org/10.5067/ASDC/SUBORBITAL/AJAX/DATA001, Iraci et al., 2021a) are readily available from NASA's Atmospheric Science Data Center (NASA ASDC, https://asdc.larc.nasa.gov/project/AJAX).

### 4.1 Long-term observations and trends

The multi-season, multi-year, multi-objective sampling approach adopted by the AJAX program allows for detailed analysis of a range of questions related to climate and air quality as well as atmospheric trends. For example, Figure 2 shows the increasing mixing ratios of $CO_2$ and $CH_4$ between 2011 and 2018 as observed at RRV, NV.

Figure 7 shows the average trace gas mixing ratios for $CO_2$, $CH_4$, and $O_3$ over the entirety of each AJAX flight (green datapoints). The average of measurements within the boundary layer (BL, defined as <2 km, teal datapoints) and averages within the free troposphere (FT, defined as >2 km, indigo datapoints) are also shown for each AJAX flight. Presented in grey are the monthly average mixing ratios from atmospheric baseline stations (Mauna Loa (MLO) for $CO_2$ and $CH_4$ and Trinidad Head (THD) for $O_3$). The mean AJAX observations of $CO_2$ (Figure 7, top panel) are in such good agreement with the surface trends at MLO that they lie directly underneath the other plotted data making them difficult to observe in Figure 7. AJAX FT $CO_2$ agrees slightly better than AJAX BL $CO_2$, which is influenced by local source emissions.

AJAX free tropospheric observations of $CH_4$ (middle panel) track the general trends in the MLO data but are offset slightly (typically less than 50 ppb). The boundary layer observations of $CH_4$ show the impact of emissions, with most flights observing large $CH_4$ AJAX BL values, relative to the FT.

$O_3$, as a secondary pollutant, is impacted by a range of processing conditions (UV, temperature, etc.) and emissions sources. Thus, most AJAX BL $O_3$ observations show larger and more variable values relative to those observed within the marine boundary layer at the THD surface site, likely indicating that AJAX BL observations are more representative of continental air masses. Detailed comparisons of AJAX to other vertically-resolved ozone observations (ozonesondes and lidar) are presented in Langford et al. (2019) and Yates et al. (2017).



Figure 8 (right) shows the relationship between $CH_4$ and $CO_2$ measured by AJAX categorized by altitude, showing that observations in the FT (representative of the well-mixed atmosphere) are more closely correlated, while the relationship between $CH_4$ and $CO_2$ in the boundary layer is complex, with a wide variety of sources resulting in a large spread of correlations. But when these data are categorized by year (Fig. 8 (left)), patterns can be seen, including the annual increase in $CO_2$ and $CH_4$. In 2011, $CO_2$ was observed between 382 ppmv and 397 ppmv, and by 2018 this range was 401-453 ppmv. The

increase in the mean annual $CO_2$ reported by AJAX was ~3.0 ppm/yr between 2011 and 2018. This estimate is in line with global growth rates of $CO_2$ reported by NOAA from observations at marine surface sites which averaged at 2.43 ppm/yr and varied between 1.90 and 3.03 ppm/yr over the same period (NOAA, 2022). Likewise, the increase in $CH_4$ between 2011 and 2018 corresponds to an increase in the mean annual $CH_4$ of ~10 ppbv/yr, which also compares well with the annual global increase of $CH_4$ reported by NOAA, which averaged at 7.69 ppb/yr and varied between 4.85 and 12.73 ppb/yr over the same

period (NOAA, 2022).

**4.2 Observations of formaldehyde and ozone in a changing boundary layer**

During the California Baseline Ozone Transport Study (CABOTS) campaign in spring and summer 2016 (Faloona et al., 2020) AJAX performed twelve flights (Iraci et al., 2020), several with multiple vertical profiles along a west-to-east transect as shown in Figure 9. This flight track provided an opportunity to assess the changes in the photochemical state of the boundary

layer over a relatively short spatial distance (~250 km) between the western-most, offshore profile and the eastern-most profile over Visalia (36.319 °N, 119.393 °W), in California's San Joaquin Valley (SJV). This helped to address one of the main aims of CABOTS, to observe the changes in $O_3$ along the coast (upwind of California) and within California's Central Valley (Faloona et al., 2020). The SJV is the lowermost part of California's Central Valley and is an area that continues to exceed health-based air quality standards for hazardous air pollutants, including $O_3$, PM 2.5 and PM 10. The ability of AJAX to

observe multiple trace gas species complemented the additional CABOTS measurement suite, including ozonesondes and ozone lidar. For example, HCHO is the third-most abundant oxygenated volatile organic compound in the SJV (Liu et al., 2022), and it undergoes complex photochemical reactions resulting in the production of $O_3$.

Figure 9 compares the west-to-east profiles for each trace gas species measured by AJAX. Each species shows a good degree of correlation at all locations in the free troposphere (>~2 km). However, in the lower troposphere each species shows

significant differences at the varied locations. For $O_3$, HCHO and $CH_4$ there is a trend to increasing values from west to east, with the smallest mixing ratios for $O_3$, HCHO, $CO_2$ and $CH_4$ observed offshore at Pt Sur, and the largest mixing ratios over Visalia (KVIS) in line with the SJV's typical spring/summer conditions (high UV, temperature) favoring photochemical processing and $O_3$ formation in the lowermost troposphere. For $H_2O$, the profile over Visalia has a similar water vapor content within the boundary layer as the offshore profile due to the presence of rivers, crop irrigation and transpiration (Cooper et al.,

280     2011).



### 4.3 Satellite validation case study: AJAX data for validation of Lowermost Tropospheric (LMT) products from OCO-2 and GOSAT

AJAX observations of $CO_2$ can be used to validate OCO-2 lowermost troposphere (LMT; approximately surface to 2.5 km, or surface to 750 hPa) and GOSAT lower troposphere (LT; defined by the retrieved surface pressure ($P_{surf}$) as 0.6-1 $P_{surf}$ or

approximately surface - 510 hPa) estimates of $CO_2$, and can, to a lesser extent, validate the OCO-2 upper column (U; approximately above 2.5 km, or 750 hPa) and GOSAT upper troposphere (UT; 0.2-0.6 $P_{surf}$) of $CO_2$.

To compare the satellite observations to aircraft, we calculate an estimate of how the satellite would observe the AJAX extended profile, given the satellite's sensitivity (averaging kernel). This is accomplished using Equation 1, where $z$ is the partial column, $x$ is the profile, $ak$ is the averaging kernel, and the $a$ subscript represents the a priori information. $z_{true,ak}$

represents the expected satellite observation of the AJAX extended profile.

$$z_{true,ak} = z_a + \sum_i ak_i * (x_{true,i} - x_{a,i}) \tag{1}$$

The OCO-2 and GOSAT partial column quantities have different sensitivities as shown in Figure 10b.

Since AJAX observations extend from near-surface to ~8-9 km, the AJAX profile needs to be extended both upward and

downward to compare to a satellite column data product. Aircraft measurements are extended down to the surface using the lowest measured value. Between the top aircraft observation and 100 hPa, we extended the AJAX profile using the CarbonTracker 2019B model (Jacobson et al., 2020) with a constant offset to match to the topmost AJAX observation. Above 100 hPa, the CarbonTracker 2019B model without offset is used. Detailed overview of the LMT and U partial column products is described by Kulawik, et al. (2017) and LT and UT by Kuze et al. (2022).

Figure 10a shows an example of a $CO_2$ profile measured by AJAX (black); the OCO-2 and GOSAT priors (green), the total column OCO-2 XCO2 value (grey), and partial column products. These data were collected on 13 April 2018 during regular nadir overpasses (looking straight down) of both satellites. The satellite observations are plotted as solid lines, with horizontal bars showing the standard deviation of all nearby satellite observations. The GOSAT upper troposphere (UT, dark blue) shows a value slightly larger than the XCO2 column, and the OCO-2 upper partial column (U, light blue) has a lower value than

XCO2. OCO-2 and GOSAT both show an enhancement in the lower troposphere (orange and red solid lines, respectively) as compared to the full column OCO-2 XCO2 (grey solid line).

AJAX data with the respective averaging kernels (using Eq. 1) are shown with dotted lines of colors matching the individual satellite partial products. The calculated LMT (orange dotted) and LT (red dotted) partial column averages are very similar to each other and to the direct observations (black). Thus, the AJAX data validate the vertical gradient observed by both GOSAT

and OCO-2, with all matches within the satellite errors (estimated by the variability of all satellite matches). Validation of

these exciting new GOSAT and OCO-2 vertically resolved products using AJAX in situ data shows the unique usefulness of aircraft profiles and the AJAX dataset of co-located vertical profiles in particular. A future paper will broaden this validation using a complete set of AJAX coincident observations over the period 2015-2018.

## 5 Data availability

The AJAX data described in this manuscript are freely available at NASA's Atmospheric Science Data Center (https://asdc.larc.nasa.gov/project/AJAX, doi: 10.5067/ASDC/SUBORBITAL/AJAX/DATA001, Iraci et al. 2021a). The data are provided in ICARTT format, which is described at https://www.earthdata.nasa.gov/esdis/esco/standards-and-practices/icartt-file-format lite product is available through NASA Goddard Earth Sciences Data and Information Services Center (GES DISC) at: https://disc.gsfc.nasa.gov/datasets/OCO2_L2_Lite_FP_10r/summary?keywords=oco-2%20r10.
GOSAT data used for this paper are available through Japan Aerospace Exploration Agency (JAXA) GOSAT/GOSAT-2 EROC daily partial column GHG website available at: https://www.eorc.jaxa.jp/GOSAT/CO2_monitor/index.html.

## 6 Summary

Measurements collected by the AJAX program provide an extensive dataset of trace gas and meteorological observations that will be of interest to a wide variety of data users. The multi-year, multi-objective, multi-instrument approach provided by the
AJAX program was unique from a NASA airborne science perspective and produced a long-term record of vertical profile (up to 8-9 km) and level leg measurements over California and Nevada. The regular sampling approach and ability to vary flight objectives based on season, synoptic and atmospheric conditions allowed for opportunistic flights and flexibility to address several science questions. The AJAX dataset is of interest to those researching areas related to (but not limited to) 1) tropospheric profiling, including satellite validation; 2) boundary layer observations in California, with an emphasis on the
San Francisco Bay Area and California's Central Valley; and 3) source identification, including wildfires, urban outflow and emissions from oil and gas and agriculture.

This paper provides a detailed overview of the AJAX program and its dataset, including a summary of flight objectives, sampling strategies, airborne instrumentation, data processing and a presentation of the overall datasets and their availability. AJAX observations can be used to validate lowermost troposphere satellite data products as shown in the case study presented
in Section 4.3.

## Author contribution

Funding acquisition, project administration and supervision were carried out by LTI. JEM, J-MR, CLP and ELY worked on data curation, formal analysis, and visualisation. SK worked on validation of OCO-2 products using AJAX data. JMS, TFH,



TPB, CC and JD-D provided resources and performed investigations. ELY prepared the manuscript with contributions from
all co-authors.

**Acknowledgements**

The NASA AJAX program recognizes support from Ames Research Center Director's funds, the NASA Postdoctoral Program
(JEM, J-MR, CLP, JT, TT, ELY), the OCO-2 Science Team, the Rapid Response and Novel research in the Earth Sciences
(RRNES) program element, and the California Air Resources Board (Contract No. 17RD004), as well as by NASA's
Atmospheric Composition Program through the Internal Scientist Funding Model and the Campaign Data Analysis and
Modeling (20-ACCDAM20-0083) program. LTI acknowledges support from the NASA Earth Science Research and Analysis
Program during data collection and analysis. TFH and JSC acknowledge support from the Goddard Internal Research and
Development (IRAD) program. Technical contributions from C. Camacho, W. Gore, P. Hamill, E. Quigley, M. Roby, J. Tadic,
T. Tanaka, T. Trias, and Z. Young made this program possible. The authors gratefully recognize the support and partnership
of H211 L.L.C., with particular thanks to K. Ambrose, T. Cardoza, R. Fisher, T. Grundherr, J. Kerr, J. Lee, B. Quiambao, R.
L. Sharma, D. Simmons and R. Simone. Data discovery, accessibility and archiving were made possible through the efforts of
the Atmospheric Science Data Center staff: K. Phillips, M. Buzanowicz, N. Jester, N. Arora, and S. Haberer. The OCO-2
lowermost tropospheric analysis was funded from ROSES 17-OCO2-17-0013 project, "Reducing the impact of model
transport error on flux estimates using CO2 profile information from OCO2 in concert with an online bias correction." JAXA-
GOSAT partial column density products were funded by JAXA GOSAT program. Authors gratefully acknowledge A. Kuze
and H. Suto of JAXA for consultation regarding the GOSAT partial column products. The TMF data used in this publication
were obtained from Dr. Thierry Leblanc as part of the Network for the Detection of Atmospheric Composition Change
(NDACC) and are available through the NDACC website www.ndacc.org.

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



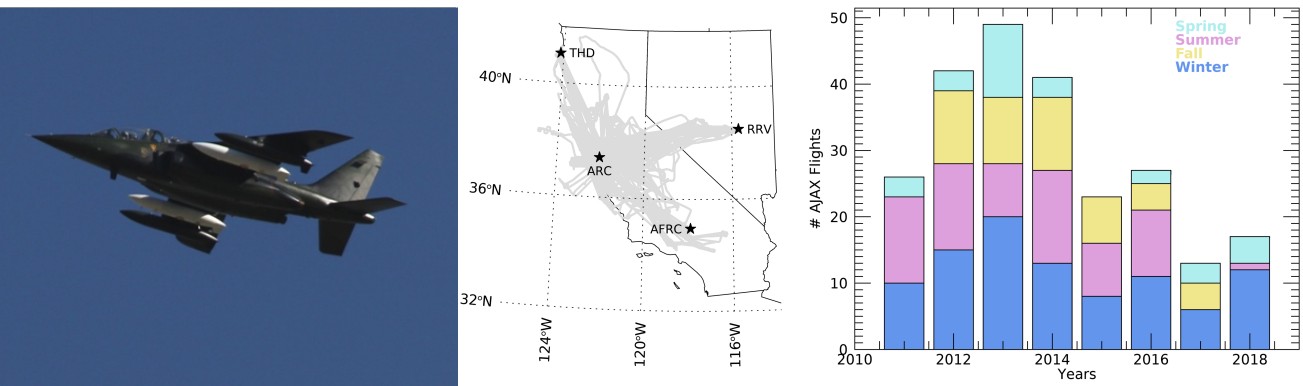

**Figure 1: AJAX in flight, before landing at Ames Research Center (ARC), located at Moffett Field, CA (left), map of AJAX flight tracks (middle), and AJAX flights by year and season (right). (AFRC = Armstrong Flight Research Center, RRV = Railroad Valley, THD = Trinidad Head).**

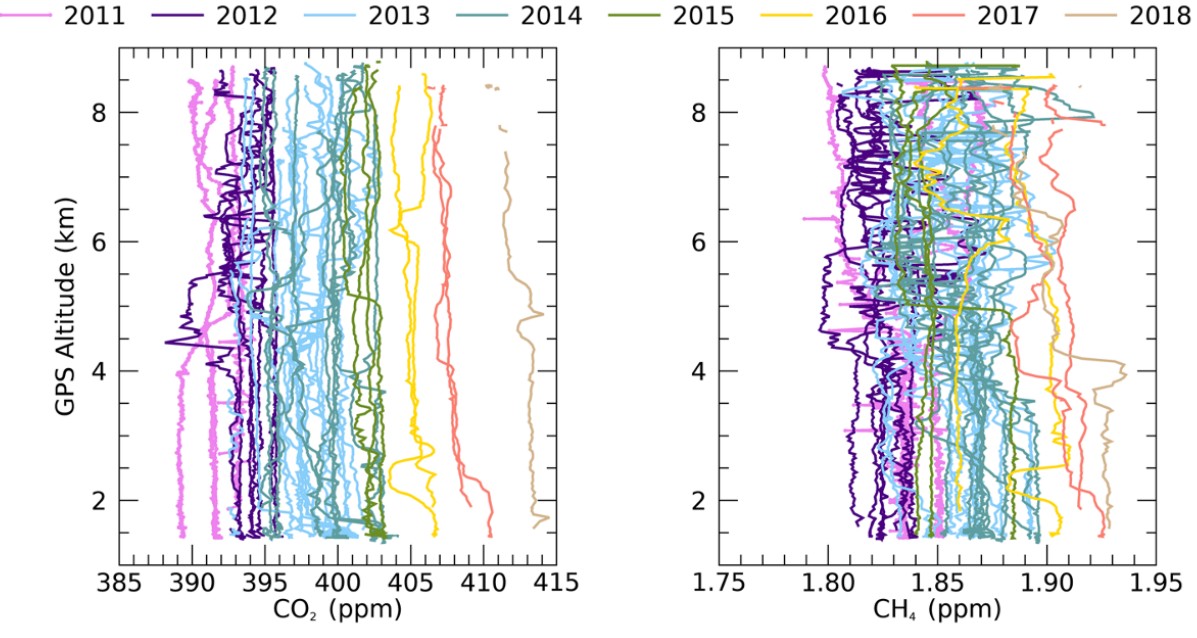


**Figure 2. AJAX observations of $CO_2$ and $CH_4$ over Railroad Valley, NV (surface elevation = 1437 m asl), colored by year of observation.**



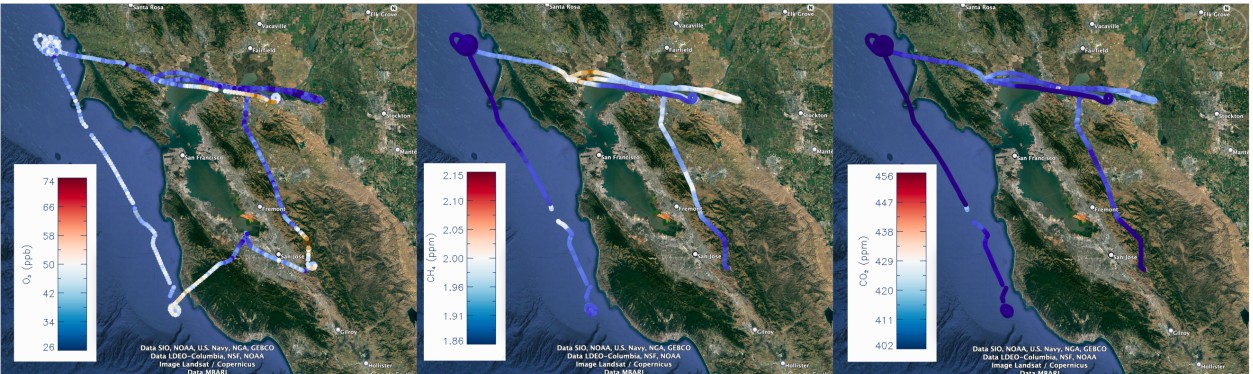

**Figure 3. AJAX Flight #202 on 2 November 2016: Observations offshore and over the San Francisco Bay Area of ozone (left), methane (middle) and $CO_2$ (right) shown in the © Google Earth map. Note that warm-up and shut down of the GHG instrument causes a loss of data on take-off (for most flights) and landing (for some flights; the post-flight procedures were modified in 2017 to retain landing data in most flights after this date).**

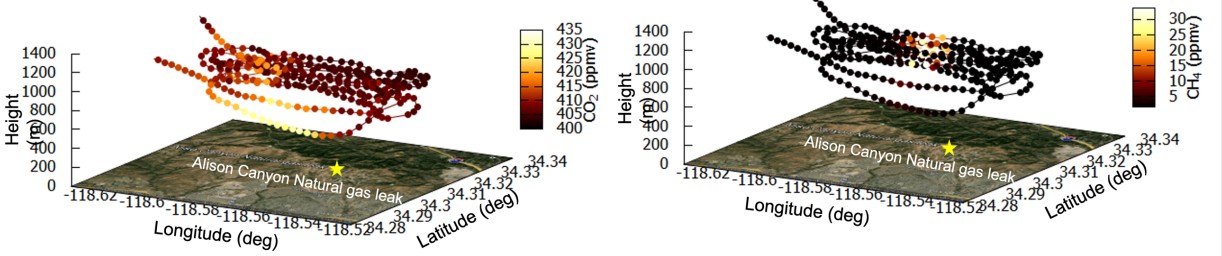

**Figure 4. AJAX Flight #175 on 4 December 2015 sampling emissions from the Aliso Canyon gas leak shown in the © Google Earth map. Maps show carbon dioxide ($CO_2$, left) and methane ($CH_4$, right) over the emission source.**

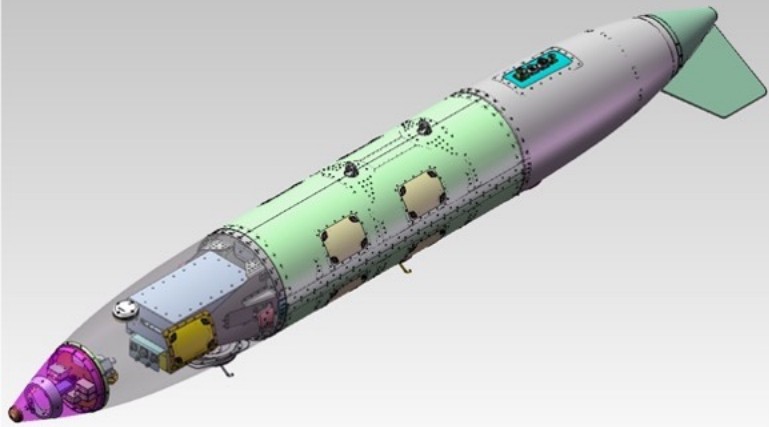

**Figure 5. CAD drawing of an AJAX wing-pod showing the location of the ozone sensor and MMS in the nose-cone section of the wing-pod. The greenhouse gas sensor is located in the middle (green) section, and its inlet can be seen below the pod. Signal and electrical connections to the aircraft are made at the top, aft panel (cyan).**

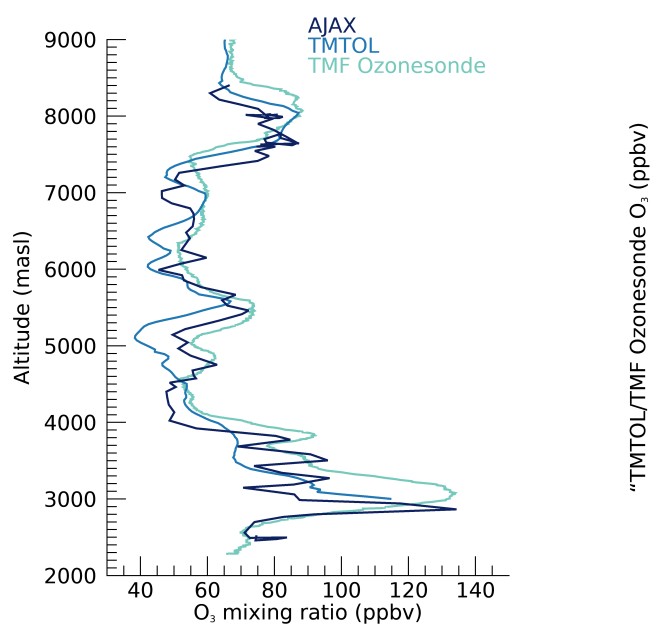
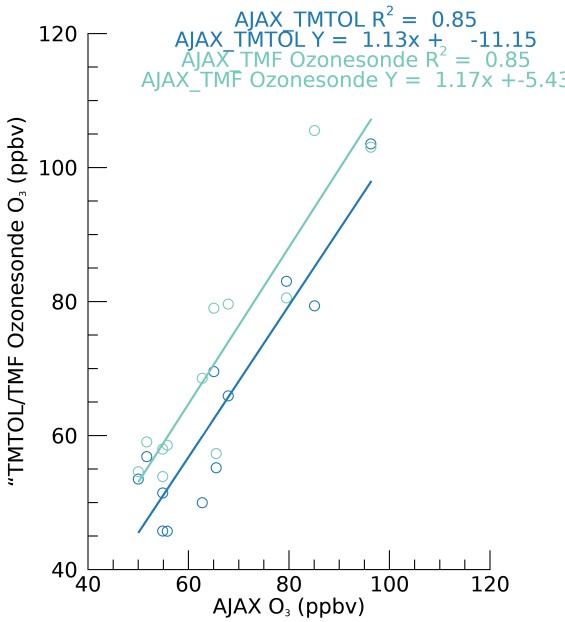

**Figure 6. Ozone profiles taken on 24 May 2013 measured by AJAX, Table Mountain Facility tropospheric ozone lidar (TMTOL) and the Table Mountain Facility (TMF) ozonesonde. Correlations of AJAX observations with TMF ozonesonde and TMTOL measurements are shown in the right panel, based on the mean ozone within 500 m vertical resolution window.**

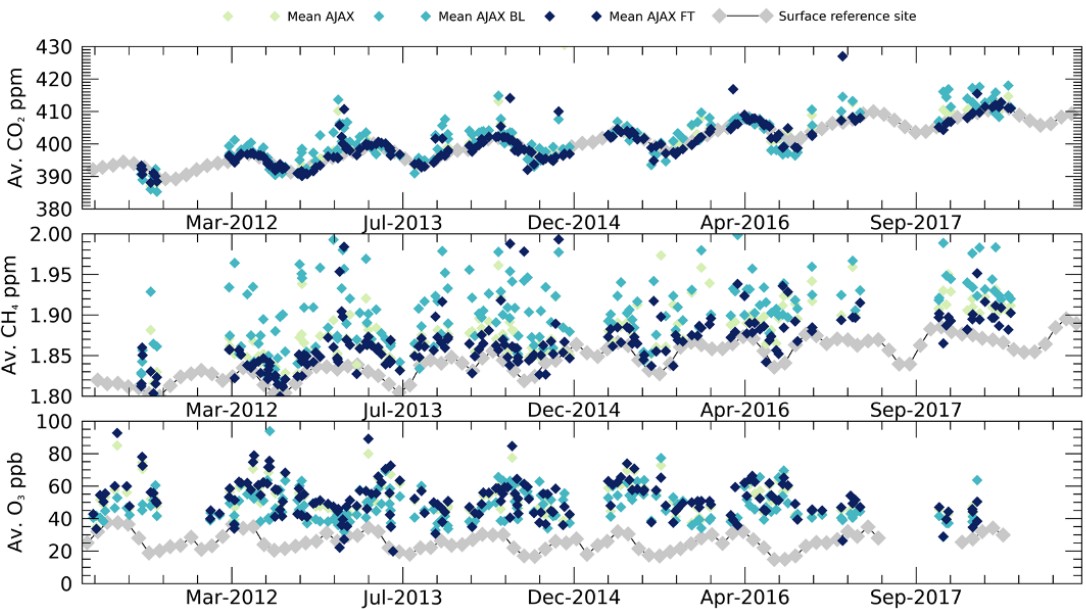

**Figure 7. Average $CO_2$ (top), $CH_4$ (middle), $O_3$ (bottom) over the entirety of each individual AJAX flight, the average of data collected below 2 km from each individual flight and above 2 km. Monthly mean values from a surface reference site are shown for comparison: Mauna Loa for $CO_2$ and $CH_4$ and Trinidad Head for $O_3$.**



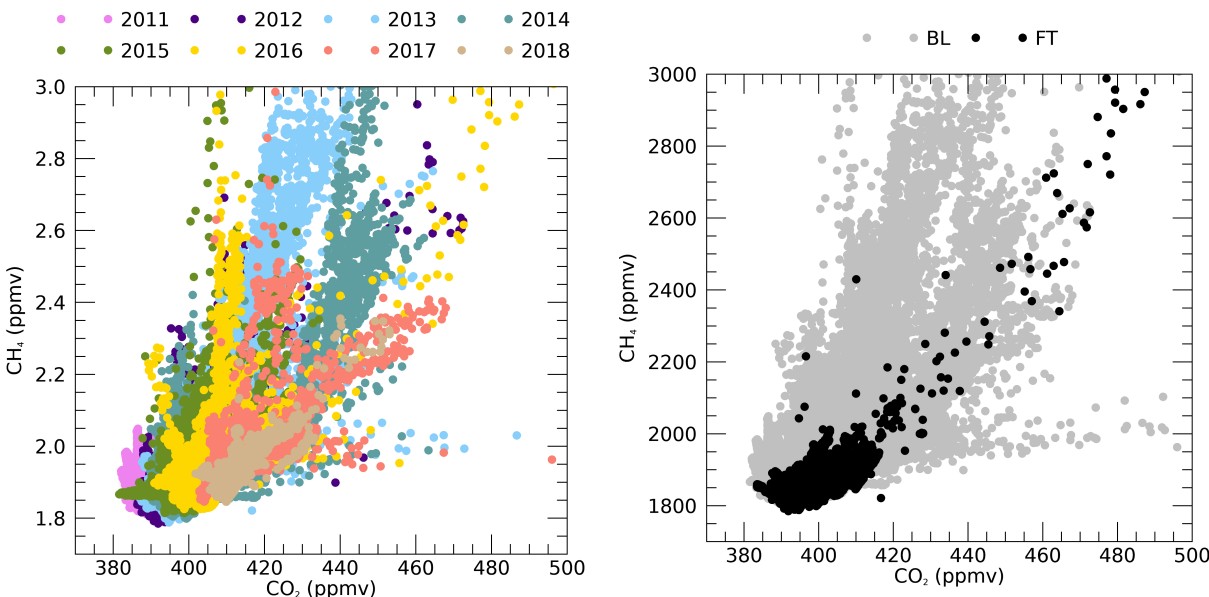

**Figure 8. Relationship between CH₄ and CO₂ measured by AJAX separated by year (left) and by free troposphere/boundary layer (FT>3 km<BL) measurements (right).**

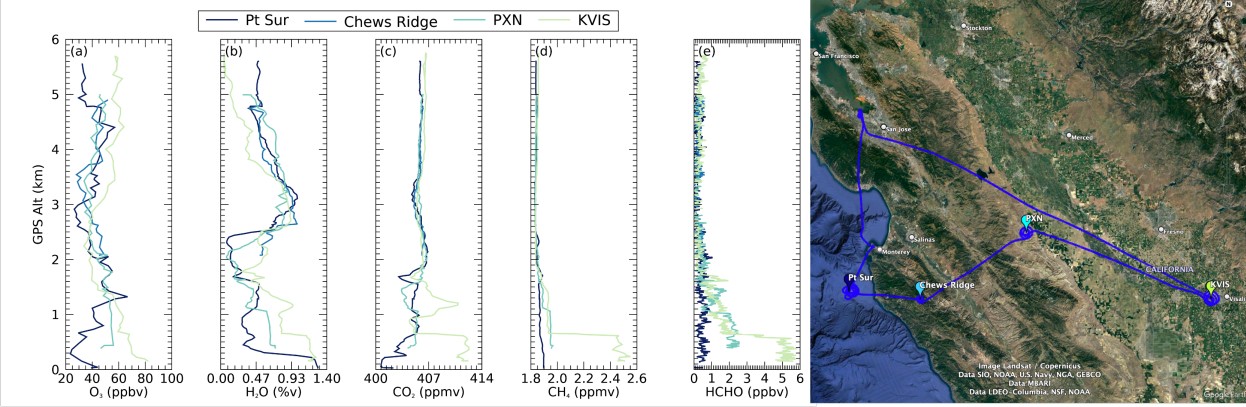

**Figure 9. Vertical profiles of O₃ (a), H₂O (b), CO₂ (c), CH₄ (d) and HCHO (e) over Pt Sur (offshore, indigo), Chews Ridge (elevation = 1521 m asl, blue), Panoche (PXN, elevation = 634 masl, turquoise) and Visalia (KVIS, green). AJAX flight track and profile locations are shown in the © Google Earth map on the right. Data are taken from AJAX flight #192 on 21 July 2016.**



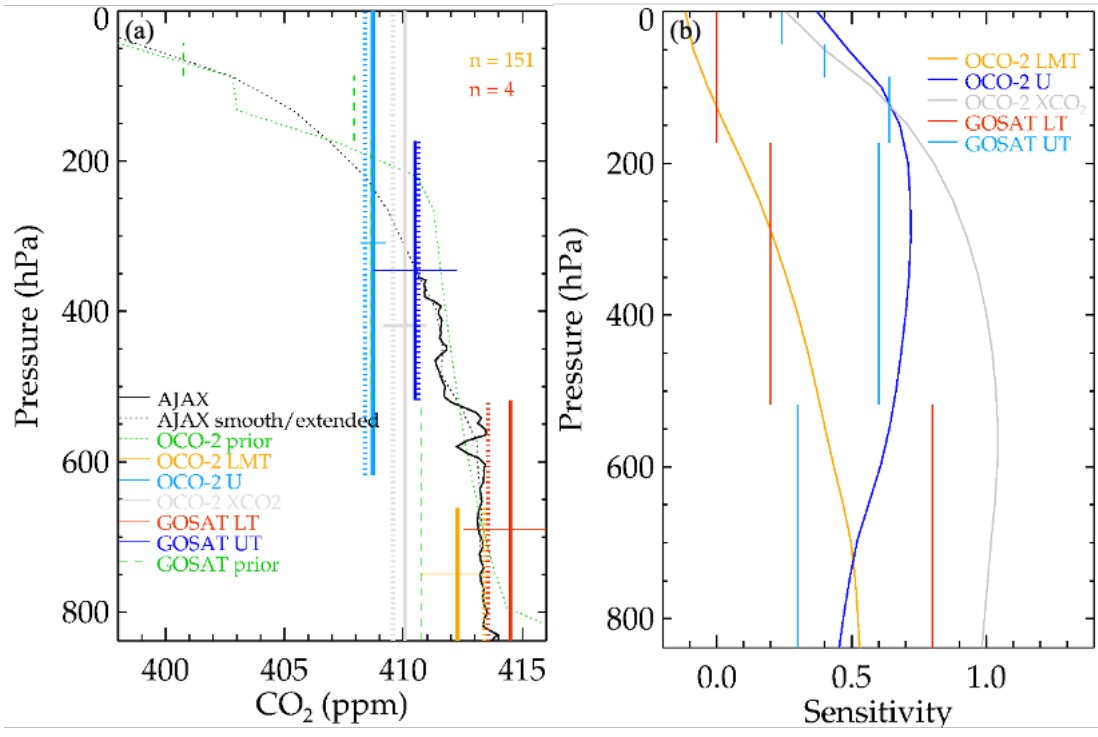

530

**Figure 10. (a) AJAX compared to OCO-2 LMT, U, and XCO2 and to GOSAT LT and UT. The black dotted line is AJAX extended by CT2019B and on the OCO-2 20-levels; AJAX data with each different averaging kernel applied (using Eq. 1) are the colored dotted lines. The OCO-2 prior is the green dotted line, and the GOSAT prior is the green dashed line. (b) Sensitivity of each satellite quantity from Fig. 10a, with the same color scheme. Note that the GOSAT averaging kernel is on 5 layers, whereas the OCO-2 averaging kernel is on 20 levels. Note that AJAX data with OCO-2 LMT (orange, dotted lines) and AJAX data with GOSAT LT (red, dotted lines) are nearly the same value, overlaid in (a).**

| Flight Objective | Description | # of flights |
|---|---|---|
| **Tropospheric profiling:** | | |
| On/Off | At least one vertical profile over land and at least one offshore (>5 km) | 40 |
| Profile | Vertical profile from ~ 8 km to surface | 127 |
| RRV | Vertical profile over Railroad Valley, NV (elev. ~1.4 km), usually coordinated with GOSAT overpass | 58 |
| TCCON | Vertical profile over TCCON instrument at NASA's Armstrong Flight Research Center (formerly Dryden) | 8 |
| THD | Vertical profile near Trinidad Head observatory, usually coordinated with NOAA ozonesonde | 11 |
| VPocean | Vertical profile over ocean, with top altitude > ~5 km | 63 |
| **Boundary layer observations:** | | |



| | | |
|---|---|---|
| CABOTS | In support of CARB's California Baseline Ozone Transport Study (Summer 2016) | 12 |
| CentralValley | Boundary layer measurements in California's Central Valley | 85 |
| DAQ | Focused on the boundary layer over Central California during DISCOVER-AQ field campaign (Jan-Feb 2013) | 6 |
| Offshore | Sampling off the California coast, does not include a vertical profile | 14 |
| SanBernardinoWestMojave | Boundary layer measurements in the San Bernardino Mountains and/or west Mojave Desert regions. | 15 |
| SFBayArea | Boundary layer measurements In the San Francisco Bay Area | 13 |
| **Source identification:** | | |
| Fire | Sampling influenced by one or more fires, usually wildfires | 17 |
| O&G | Sampling over known oil and/or gas infrastructure | 17 |
| Urban Outflow | Boundary layer measurements around Sacramento, CA | 12 |

**Table 1: Definition and description of common AJAX flight objectives and count of flights that contain each objective. Note that many AJAX flights accomplished multiple objectives in a single flight.**

| Error source (10s sampling resolution) | Ozone (ppb) |
|---|---|
| Laboratory precision | 1.2 |
| Variability in zero offset | 1.4 |
| Repeatability | 0.6 |
| Pressure dependance | 2.2 |
| Overall uncertainty (RMS of 1-σ uncertainties) | 3 |

540 **Table 2. Uncertainty of the AJAX $O_3$ measurements.**

| Error source (3s sampling resolution) | CO₂ (ppm) | CH₄ (ppb) |
|---|---|---|
| Laboratory precision | 0.03 | 0.3 |
| Accuracy of the standard | 0.07 | 0.3 |
| Variance in flight | 0.08 | 0.8 |
| Uncertainty due to water vapor corrections | 0.10 | 2.0 |
| Repeatability | 0.03 | 0.4 |
| Pressure dependance | 0.04 | 0.1 |
| Overall uncertainty (RMS of 1-σ uncertainties) | 0.16 | 2.2 |

**Table 3. Uncertainty of the AJAX $CO_2$ and $CH_4$ measurements.**