# Peer review of "An extensive database of airborne trace gas and meteorological observations from the Alpha Jet Atmospheric eXperiment (AJAX)"

_Earth System Science Data, 2023_

## Referee Comment (RC2)

**Review comments for "An extensive database of airborne trace gas and meteorological observations from the Alpha Jet Atmospheric eXperiment (AJAX)" by Emma L. Yates et al.**

This manuscript presents a multi-year dataset from the AJAX airborne observations of tropospheric ozone, $CO_2$, $CH_4$, and HCHO, together with meteorological parameters over an extended area of California. These data provide valuable information for a broad range of research, as demonstrated by cited publications, making this work highly relevant for publishing in the journal ESSD. Overall, the manuscript is well-organized and well-written. However, there are a few gaps and issues that need to be addressed. Please see my suggestions below.

1) Inception of the project and the design of the project scope

The introduction briefly mentions that AJAX is a project of a government-run laboratory partnered with a private company and is considered a project of "opportunity." In my experience, this is an unusual model for an airborne atmospheric experiment. As a project and data overview, it would be important for the readers to see a brief description of the project formation and the concept of the payload design, which must be related to intended scientific applications. Currently, the manuscript focuses on the number of flights and instruments that produced the multi-year, multi-objective dataset without explaining the motivations and scientific scope at the project's inception. Providing a brief conceptual and design introduction would help the multi-objective flights and data description take on an active voice, giving the reader a better context of the data.

2) Information flow

There are several information flow issues that created repetitions. An outstanding example is the paragraph starting at line 225 that overlaps with section 5 data availability.

Try to introduce the platform (range and ceiling) and the payload (variable measured) before sampling strategy. It may work better.

3) Issues with Section 4 "dataset overview"

This is a problematic section. It is possible to integrate the first paragraph (line 225) with section 5 and called it "Dataset overview and availability".

The remaining contents of section 4 do not provide an overview, but rather showcase the dataset's application through three examples. Although these examples effectively highlight the rich information contained in the dataset, the style of the discussions in this section needs improvement. The main issue is that it is unclear whether these examples are written as mini data analysis papers, each leading to conclusions, or merely as a conceptual demonstration of possible applications in these problems. Assuming that your intention is the latter, it is acceptable to use a hand-waving style of writing. However, you should (a) clearly state this

intention and (b) provide references to any concepts or methods used but not sufficiently introduced in the text.

Specific examples:

- In Section 4.1, the data application in trend analysis is showcased through figures that clearly display the available information in the dataset. However, the statement "The increase in the mean annual CO2 reported by AJAX was ~3.0 ppm/yr between 2011 and 2018" is not supported by a reference or a detailed explanation of how the result was derived.

- Section 4.3 would benefit from some simplifications and more referencing. Although the method described is well-established, it is problematic without a couple of references here for the concept behind eq (1). Since this section overlaps the topic of 2.1, I wonder if you already have a publication for this work, where eq (1) is better introduced. Otherwise, it is appropriate to cite a paper that introduced the method, possibly go back to Rogers & Connor (2003). Additionally, XCO2 should be defined to ensure clarity for readers who may be unfamiliar with the term.

4) Some editorial comments, primarily on figure clarity

Most of the issues are related to the color choices in the figures, as they are not suitable for printing. I would recommend printing a copy and reviewing it to identify areas where the color choice needs to be changed.

- Line 26: "modified for  …"
- Fig. 1 (left): reduce the background sky color to a much lighter blue, so the plane is visible in print.
- Fig. 3: change color table for the track to use lighter and brighter colors to avoid blue on blue.
- Fig. 9 (right): This map is not readable in print – change track color and increase the fonts of labels. Yes, you can do that in Google Earth.
- Fig. 10 (left) the dotted lines are too faint.

---

## Author Comment (AC1)

The authors would like to thank the referees for their reviews and comments; we feel the revised manuscript has been significantly improved as a result. We have responded to each comment separately below, our response is in *italics.*

**Reviewer #1:**

In their manuscript, the authors present results from a large number of scientific flights within the AJAX program. In a first part, the program itself is motivated and examples of scientific objectives for some of the AJAX flights are given, together with operational aspects of the program. Then, individual instruments are briefly introduced and generated data sets are characterized. Finally, examples for the usefulness of AJAX data are given: Long-term trend observations (and comparisons to ground based measurements), observations of pollutants in the boundary layer, and satellite validation.

The manuscript is well written, figures are of high quality and the analyses the authors did are sound. The paper is well within the scope of ESSD and it is in a very good shape. I only have a few minor points, which could be addressed by the authors before publication.

*Response: Thank you for your comments. We have carefully addressed each point and incorporated them in the revised manuscript.*

Minor/specific points:

- line 37: It might be helpful to the reader to give some references for the mentioned "traditional" field campaigns. For my taste, it would not be necessary to resolve the acronyms in this context, but some idea where to find context information would be nice.

*Response: Done*

- line 43: Please also give metrical units (meters or kilometers) for the altitudes, even though elevations in "feet" are usual in aviation.

*Response: Done*

- line 81: suggestion: (Figure 1) -> (Figure 1, middle panel)

*Response: Done*

- Figure 1: the right panel with the time series of the flights is very informative, but not introduced in the main text. Please refer to it somewhere, e.g. in the beginning of section 2.

*Response: We have added this reference to the end of the 1$^{st}$ paragraph in the introduction section.*

- line 99: I suggest to remove the specific FAR number in the text and just refer to regulations and give the reference as it is given now.

*Response: Done*

- line 125: Maybe not everyone is familiar with the location of Aliso Canyon, so maybe a small hint, where it is located would be helpful.

*Response: Sentence now reads: "An example source identification AJAX flight is shown in Figure 4 from a flight over the Aliso Canyon, CA natural gas leak in December 2015 (34.315 °N, 118.564 °W)."*

- Figure 4: For both panels, the label of the z-axis is displayed in two lines, which overlaps with the ticks of the axis. Sometimes it is impossible to make the plotting program do the labels in a beautiful way, but if possible, I would suggest to have the label in only one line.

*Response: Done*

- line 141: I think the common abbreviation of kilometers per hour is "km/h" (without "r")

*Response: Done*

- Sections 3.1 and 3.2: I miss the measurement principle of these two kind of instruments. In my opinion, it is not necessary to explain the instrument in detail in this manuscript, since references are given, but I would at least like to read the name of the technique instead of just knowing that it is a "sensor".

*Response: The measurement principle has been added to the introductory lines of each section.*

- line 220: mb -> hPa

*Response: Done*

- Sections 3.3 and 3.4: For the O3 and GHG instruments, a detailed procedure of data screening has been reported, which is not presented for the HCHO and MMS instruments. Is there no data screening for these instruments?

*Response: We have added the filtering steps for final HCHO data to section 2.3 and additional information for MMS in section 2.4. (Note we have re-ordered the sections so that the instrumental section is now section 2 as suggested by reviewer 2).*

- Figure 7: The legend of this figure has very small fonts and is hard to read. In addition, the light green color is hard to see in the plots. Further, I suggest to mention the colors in the figure caption, e.g.: "Average CO2 (top), CH4 (middle), O3 (bottom) over the entirety of each individual AJAX flight (green), the average of data collected below 2 km from each individual flight (BL, cyan), and above 2 km (FT, dark blue). Monthly mean values from a surface reference site  are shown for comparison (grey): ..."

*Response: We have increased the font size for the legend and changed the color scheme for the figure to color brewer purples scheme (lighter to darker shades) which we believe makes the figure easier to read while meeting the journal requirements for accessibility of color figures.*

- Section 4.1: I think it would be good to add references for the ground based measurements, the AJAX data are compared to.

*Response: Done*

- line 247: Just out of curiosity: Are the AJAX data points matching better to the THD curve those flights, which have been above the pacific ocean (as shown in the map in Fig. 1)?

*Response: In the below figure, we filtered the AJAX O3 data to calculate the mean AJAX O3 based only on datapoints with a longitude less than -121W, which corelates to mostly offshore O3 datapoints. Comparing this figure to Figure 7, shows that filtering based on "offshore only" does not change the observations relative to the Trinidad Head surface site. Yates et al., (2017) provides a more detailed comparison of AJAX data and Trinidad Head (as well as the TOPAZ lidar at the Table Mountain Facility).*

[Figure]

- line 292: This one-sentence-paragraph feels a bit odd here and should be extended a little. It is also strange that Fig. 10b is introduced before Fig. 10a. However, I am not really sure, if Fig. 10b is really necessary in this work at all (but it would be okay if the authors want to keep it).

*Response: We have incorporated this sentence into the paragraph above. We switched 10a/left and 10b/right. (now) 10left (sensitivities) is useful to show the difference in the sensitivities between the satellites and how Eq. 1 is used to compare the satellite and airplane observations.*

- Figure 10a: I am not sure, if I understand the meaning of the "n = ..." statements in this panel.

*Response: Updated caption to add, "The n values are the corresponding number of satellite observations that are with the coincidence criteria, and averaged together, where the error bars shown correspond to the standard deviation of the satellite values."*

**Reviewer #2:**

This manuscript presents a multi-year dataset from the AJAX airborne observations of tropospheric ozone, $CO_2$, $CH_4$, and HCHO, together with meteorological parameters over an extended area of California. These data provide valuable information for a broad range of research, as demonstrated by cited publications, making this work highly relevant for publishing in the journal ESSD. Overall, the manuscript is well-organized and well-written. However, there are a few gaps and issues that need to be addressed. Please see my suggestions below.

*Response: Thank you for your comments. We have addressed each point in the revised manuscript.*

1) Inception of the project and the design of the project scope

The introduction briefly mentions that AJAX is a project of a government-run laboratory partnered with a private company and is considered a project of "opportunity." In my experience, this is an unusual model for an airborne atmospheric experiment. As a project and data overview, it would be important for the readers to see a brief description of the project formation and the concept of the payload design, which must be related to intended scien0fic applications. Currently, the manuscript focuses on the number of flights and instruments that produced the multi-year, multi-objective dataset without explaining the motivations and scientific scope at the project's inception. Providing a brief conceptual and design introduction would help the multi-objective flights and data description take on an active voice, giving the reader a better context of the data.

*Response: Thank you. We have added a brief description of the project concept and design in the first paragraph of the introduction which helps to explain the "unusual" model of the AJAX program.*

2) Information flow

There are several information flow issues that created repetitions. An outstanding example is the paragraph starting at line 225 that overlaps with section 5 data availability.

*Response*: *Thank you for this suggestion. We have combined the paragraphs in Section 5.*

Try to introduce the platform (range and ceiling) and the payload (variable measured) before sampling strategy. It may work better.

*Response: We have re-structed the paper as suggested to describe the platform and sensors prior to the sampling strategy, this does indeed help with the flow of the paper, thank you.*

3) Issues with Section 4 "dataset overview"

This is a problematic section. It is possible to integrate the first paragraph (line 225) with section 5 and called it "Dataset overview and availability".

*Response: Thank you for this suggestion. We have combined the paragraphs in Section 5.*

The remaining contents of section 4 do not provide an overview, but rather showcase the dataset's application through three examples. Although these examples effectively highlight the rich information contained in the dataset, the style of the discussions in this section needs improvement. The main issue is that it is unclear whether these examples are written as mini data analysis papers, each leading to conclusions, or merely as a conceptual demonstration of possible applications in these problems. Assuming that your intention is the latter, it is acceptable to use a hand-waving style of writing. However, you should (a) clearly state this intention and (b) provide references to any concepts or methods used but not sufficiently introduced in the text.

*Response: You are correct our intention was the latter; this section is meant to highlight the data and its applications to various studies. We have edited Section 4 to include introductory statement clearly explaining this intention.*

Specific examples:

- In Section 4.1, the data application in trend analysis is showcased through figures that clearly display the available information in the dataset. However, the statement "The increase in the mean annual CO2 reported by AJAX was ~3.0 ppm/yr between 2011 and 2018" is not supported by a reference or a detailed explanation of how the result was derived.

*Response: This value was calculated by firstly taking the mean AJAX CO2 value for each year then calculating the year-to-year difference (i.e. mean 2012 CO2 – mean 2011 CO2). Then taking an average of the yearly increases observed over 2011-2018 which was 3 ppm/yr. Which is in-line with reported global growth rates reported by NOAA. We have re-worded the sentence to help explain this better.*

- Section 4.3 would benefit from some simplifications and more referencing. Although the method described is well-established, it is problematic without a couple of references here for the concept behind eq (1). Since this section overlaps the topic of 2.1, I wonder if you already have a publication for this work, where eq (1) is better introduced. Otherwise, it is appropriate to cite a paper that introduced the method, possibly go back to Rogers & Connor (2003). Additionally, XCO2 should be defined to ensure clarity for readers who may be unfamiliar with the term.

*Response: We have made changes to the text in Section 4.3 and including additional references and definition of XCO2. We have cited Rodgers, 2000; Conner et al., 2008; Kulawik et al., 2017.*

4) Some editorial comments, primarily on figure clarity

Most of the issues are related to the color choices in the figures, as they are not suitable for printing. I would recommend printing a copy and reviewing it to identify areas where the color choice needs to be changed.

*Response: We did as suggested and made changes to some color schemes, choices and line thickness etc. which make the figures more suitable.*

- Line 26: "modified for for ..."

*Response: Corrected*

- Fig. 1 (left): reduce the background sky color to a much lighter blue, so the plane is visible in print.

*Response: Done.*

- Fig. 3: change color table for the track to use lighter and brighter colors to avoid blue on blue.

*Response: Done.*

- Fig. 9 (right): This map is not readable in print – change track color and increase the fonts of labels. Yes, you can do that in Google Earth.

*Response: Done.*

- Fig. 10 (left) the dotted lines are too faint.

*Response:  Made dotted lines thicker in Fig. 10.*